# Effect of Physiotherapy Treatment with Immersive Virtual Reality in Subjects with Stroke: A Protocol for a Randomized Controlled Trial

**DOI:** 10.3390/healthcare11091335

**Published:** 2023-05-05

**Authors:** Aitor Garay-Sánchez, Yolanda Marcén-Román, Mercedes Ferrando-Margelí, M. Ángeles Franco-Sierra, Carmen Suarez-Serrano

**Affiliations:** 1Institute for Health Research Aragón, 50009 Zaragoza, Spain; 2Miguel Servet University Hospital, 50009 Zaragoza, Spain; 3Department of Human Anatomy and Histology, Faculty of Medicine, University of Zaragoza, 50009 Zaragoza, Spain; 4Department of Nursing, Physiotherapy and Occupational Therapy, Faculty of Health Sciences, University of Zaragoza, 50009 Zaragoza, Spain; 5Department of Physiotherapy, Faculty of Nursing, Physiotherapy and Podiatry, University of Seville, 41009 Seville, Spain

**Keywords:** stroke, static balance, dynamic balance, immersive virtual reality, exercise, physiotherapy modalities

## Abstract

Background: Many stroke survivors suffer from sensorimotor deficits, especially balance impairments. The purpose of this trial is to investigate whether the designed Immersive Virtual Reality training program is better in the short term (15 sessions) and in the medium term (30 sessions) than physiotherapy training with Bayouk, Boucher and Leroux exercises, with respect to static balance in sitting and standing, dynamic balance and quality of life in patients with balance impairment in stroke survivors. Methods: This study is a randomized controlled trial with two treatment arms and evaluators blinded, and a functionality treatment group in combination with specific balance exercise training according to Bayouk, Boucher and Leroux (control group) or a balanced treatment using Immersive VR. The primary outcome will be static, Dynamic balance and gait measured by Bestest Assessment Score (BESTest), Berg Scale (BBS), Pass Scale (PASS) and Time Up and Go test (TUG). The secondary outcome will be the stroke-associated quality of life using the Stroke Quality of Life Scale (ECVI-38). Conclusions: The results of this study may add new insights into how to address balance using Immersive Virtual Reality after a stroke. If the new training approach proves effective, the results may provide insight into how to design more comprehensive protocols in the future for people with balance impairments after stroke.

## 1. Introduction

Worldwide, stroke is considered the second leading cause of death, and the third of death and disability combined. Although advances in treatment encourage a decrease in mortality among subjects who have had a stroke, among survivors, there is a great functional impairment, which is one of the main causes of disability, due to the persistence of symptoms in the long term [1,2,3].

Cerebrovascular accident (CVA) or stroke causes a sudden disruption of physiological brain function leading to impairment of functional brain networks [4]. The alteration of balance in stroke that occurs after having suffered brain injuries affects mainly the parieto-temporal or parieto-vestibular areas. This situation can cause a significant problem for the independence of these patients since they present a loss of the sense of touch, protective reactions and proprioceptive sense; all these losses are correlated with the ability to balance, which will affect mobility, and it is the main consequence of falls in subjects with Ictus [5].

Through sensory integration, the brain organizes somatosensory, visual and vestibular information and provides crucial information that is used for complex motor skills (maintaining balance and walking interacting with the environment) [6,7]. This sensory integration could be the key to promoting recovery through repetitive goal-oriented intensive therapeutic interventions and appropriate non-invasive brain stimulation [8].

Physiotherapy intervention programs in patients post-stroke must develop strategies to assess functional deficit, prevent maladaptive plasticity and maximize functional gain. For relearning and functional training, the intervention activities require motor control and must comply with the following principles: movements close to normal, muscle activation, movement conduction, focused attention, repetition of desired movements, training specificity, intensity and transfer [9,10,11]. The most widely used conventional physiotherapy intervention programs in the hospital setting are based on these principles.

Studies show that the combination of conventional physiotherapy intervention protocols with the use of systems of training through Virtual Reality (VR) optimizes the results in the recovery of the functional deficit of patients post-stroke [12,13,14,15,16,17].

Most of the systems included in the current literature are based on the use of video games or fictitious digital environments, such as Non-Immersive Virtual Reality, where the sensory inputs received by the patient do not correspond to reality, a situation that makes it difficult to adequate sensory integration [18,19,20,21].

In the absence of clarity in the use of VR terminology, we find great differences between systems labeled VR and the level of immersion, the extent to which virtual information is mixed with that of the real world and the type of input devices used [22].

A non-immersive virtual environment is commonly experienced in two dimensions, and subjects can interact via an avatar [23] (allocentric perspective) with the environment displayed on a screen [24].

In contrast, in an immersive virtual environment, subjects could egocentrically operate in a simulated world environment [25], allowing interaction with a visual perspective in real-time based on the position of the body/head, where the subject himself is who interacts with the scenarios and proposed tasks.

On the other hand, there are few previous studies using Immersive Virtual Reality in general; this may be because non-immersive video game systems developed for entertainment are less expensive, which has made this modality more accessible for possible rehabilitation interventions in stroke patients [26]. If we focus on Immersive VR, it has recently started being used as a tool to allow people with sensorimotor deficits to practice challenging skills in a safe environment [17,20,21,26,27,28,29]. 

There are studies that have demonstrated the effectiveness of the application of this immersive technology in older adult patients with cognitive impairments, where patients had moderate emotional learning and showed greater confidence in multisensory integration for learning [27,30,31,32], and also in Parkinson’s disease [28,33,34,35] and on the balance, mobility and fatigue in multiple sclerosis [29,36]. In addition, the use of gamified Immersive Virtual Reality adapted to the patient obtained good results in the rehabilitation of neglect symptoms after a stroke. [37]. Neural activity has also been seen to increase after Immersive VR intervention, particularly in brain areas involving activated mirror neurons, as in the primary motor cortex [27,31,32,38].

This study aims to determine whether the designed Immersive VR training program is better in the short term (15 sessions) and medium term (30 sessions) than physiotherapy training with exercises by Bayouk, Boucher and Leroux [39] regarding the change of measures related to static sitting, and standing balance and dynamic balance in post-stroke patients.

Secondary objectives verify the impact on quality of life and safety of the application of this type of technology.

## 2. Materials and Methods

### 2.1. Study Design and Setting

The RCT is a parallel-group design, with 1:1 randomization to a designed Immersive VR program related to the improvement of motor control during static, dynamic balance and gait in patients post-stroke, compared with usual physiotherapy care intervention, applying exercises for static and dynamic balance according to Bayouk, Boucher and Leroux [39]. This protocol follows the SPIRIT 2013 [40] recommendations for clinical trial protocols Identifier NCT04379687. The study flow is outlined in Figure 1.

This study is conducted at the Physiotherapy Unit of the Miguel Servet University Hospital in Zaragoza.

### 2.2. Participants

Eligible patients who agree to participate will meet the following inclusion criteria: Adults older than 18 and younger than 80 years old, with a diagnosis of stroke with hemiparesis or hemiplegia post-acute stroke admitted to hospital, and with a minimum score of 2 points in item 3.2 of the BBS (Berg Balance Scale) [41], which establishes that the patient can remain seated for 30 s without help. Exclusion criteria include aphasia, scores greater than 45 on the Mississippi Aphasia Screening Test [42]; cerebellar pathology; hemineglect or previous neurological disorder; visual disorders that prevent the use of VR glasses; moderate cognitive impairment, scores less than 24 on the MMSE (Mini-mental State Examination) [43]; previous musculoskeletal disorders that hinder or prevent balance in sitting and standing or gait; vestibulocochlear disease; spinal cord injury or disease; systemic peripheral neuropathy; persistent phobic postural dizziness; migraine; Ramsay Hunt Syndrome; and psychiatric disorders. 

### 2.3. Randomization of Subjects

Eligible patients recruited by nursing will be randomized as they are referred for physiotherapy treatment by the supervisor of the Physiotherapy Unit to one of the two trial groups after signing the informed consent. The sequence will be performed randomly in a 1:1 allocation to the Immersive VR training group and the Bayouk, Boucher and Leroux physical therapy training group for balance, using a computerized random allocation program.

This sequence will be prepared by an investigator not involved in the allocation of patients. 

### 2.4. Intervention Protocol/Procedure

Eligible subjects will be identified by the stroke diagnosis; they will be patients admitted to the Neurology Service of the Miguel Servet Hospital in Zaragoza. In an initial visit, the nursing staff will inform those potentially eligible, who will be pre-selected based on whether or not they meet the inclusion and exclusion criteria for this study. Likewise, they will proceed to explain this study to the included subjects, collect the willingness to participate and provide the informed consent document.

Prior to random assignment, two physiotherapists who are experts in neurological assessment will evaluate the outcome variables for each participant. Subsequently, patients will be randomly assigned to the control group or the experimental group in a 1:1 ratio, computerized random assignment program. The patient will be informed that in subsequent evaluations, they do not reveal to the evaluators to which group they have been assigned.

### 2.5. Physiotherapists Training

The interventions in both groups will be carried out by 4 physiotherapists with at least 5 years of experience in patients post-stroke management. In addition, physiotherapists will attend a 2 h training session to improve the standardization of interventions and receive a detailed procedure for each intervention to ensure they are managing patients according to trial protocols as well as recording adverse events if required. 

### 2.6. Intervention

Table 1 shows the treatment for each patient, which will be 30 min, 5 days a week.

The first part of the treatment, lasting 15 min, will be common to both groups and will be focused on the function of the structures; the following exercises will be prescribed for the flexor muscles of the upper limbs and the extensor muscle group of the lower limbs [44]:-Stretching exercises lasting 30 s each, with a maximum duration of 5 min;-Joint mobilization exercises of the hip, knee and ankle of both lower extremities, which will be carried out passively and with a total duration of 5 min;-Exercises for the active mobilization of the spine (flexion, extension, turning, etc.) with a maximum duration of 5 min.

The design of the second part of the treatment protocol will depend on whether the subjects are assigned to the control group or the Immersive VR intervention group (experimental group).

In the control group, this part of the treatment will be based on a static balance training program that has demonstrated its effectiveness; therefore, some of the exercises proposed by the authors Bayouk, Boucher and Leroux in their study have been used [39]. The exercises included in the intervention and their progression will depend on the initial situation of the participant. These exercises are shown in Table 2 and must be performed with opened and closed eyes on a hard and soft surface. 

In the experimental group, this second part will be a training program of static and dynamic balance in sitting and standing through Immersive VR. The application of Immersive VR is carried out through a Google device that has an application installed on the “FisioVR” device (Abaco Digital, Zaragoza, Spain), with dynamic and static experiences associated with different postural tasks in a realistic 360° environment. It is based on the need for people to receive treatment with the greatest freedom of movement that this device provides, as they are connected to the computer via Wi-Fi. Scenes are integrated in which they can start both standing and sitting in urban environments performed by photogrammetry and 360-degree spherical imaging, and, most importantly, it allows the patient to move and explore the environment.

The designed scenarios recreate realistic environments for their recovery (Figure 2a,b). The patient must interact in the different sessions with 5 proposed scenarios (Figure 3). In these scenarios, they must solve double-task situations. On the one hand, the postural task using different positions in sitting, standing or walking, and on the other, the cognitive task using different elements within the scenarios (visual, arithmetic and auditory). Patients will progress in difficulty as their functional capacity allows, beginning with the sitting position and ending with ambulation. At the end of the sessions, they must go through all the positions, holding each position for 5 min.

### 2.7. Outcomes

All study participants will be evaluated before starting treatment (baseline) at 15 sessions and 30 sessions. The outcome measures are compiled in Table 3. 

The parameters of balance in sitting and standing static and dynamic and in the temporospatial and kinematic parameters of gait will be obtained from the following tests and scales. 

For static balance and functional mobility, the PASS Spanish version [21,45], the BBS [34] and the BESTtest [46] have been used. 

BBS has a high relative reliability with interrater reliability estimated at 0.97 (95% CI 0.96 to 0.98) and intra-rater reliability estimated at 0.98 (95% CI 0.97 to 0.99). The absolute reliability of the BBS varies across the scale, with minimal detectable change with 95% confidence varying between 2.8/56 and 6.6/56 [41].

PASS is a 12 items performance-based scale used for assessing and monitoring postural control following a stroke. It consists of a 4-point scale where three items are scored from 0 to 3, and the total scoring ranges from 0 to 36 [45]. The minimal detectable change in subacute stroke is 2.22 points [47]; in acute stroke, 1.8 points [48]; and in chronic stroke, it is 3.2 points [49].

BESTest consists of 27 tasks, with 36 items in total. Each item is scored based on ordinal scale scoring from 0 to 3, where 3 represents best performances, and 0 represents worst performances. Scores for the total test are provided as a percentage of total points [50]. Additionally, the minimal detectable change in subacute stroke is 7.81 points [50]. 

To measure dynamic balance and gait, the TUG test [51] and the 10 m walk test [52] will be used. 

In the TUG test, the patient rises from a chair, walks three meters in a straight line, turns, returns and sits down while the examiner times the time required to perform the test. In addition, the TUG assesses the risk of falling. A score of less than 10 s indicates a low risk of falling, between 10 and 20 s indicates frailty with a higher risk of falling, and more than 20 s a high risk of falling [51]. The minimum detectable change in stroke is estimated at 2.9 s [53].

10 m walk test has an excellent interrater reliability estimated ICC = 0.87 to 0.88, and an excellent interrater reliability ICC = 0.998. The minimum detectable change is set at 0.05m/s, and a substantial change is set at 0.10m/s [52].

The secondary outcomes will be measured with the following tools: quality of life with the ECVI-38 [54] and the adverse effects with a notebook for recording if they happen derived from the treatment.

### 2.8. Participant Timeline

The schedule of enrollment, interventions and assessments of this study is shown in Table 4, following the SPIRIT statement [40].

### 2.9. Blinding

Assessments regarding clinical recovery will be conducted by two evaluators blinded to treatment assignment. Due to the nature of the intervention, participants cannot be blinded to treatment assignment. The physiotherapist performing the intervention will also not be blinded but will be asked not to disclose the patient’s assignment status at any time or during subsequent assessments. An analyst outside the research team will enter the data into the computer on separate data sheets so that the investigators can analyze the data without having access to the allocation information.

### 2.10. Data Collection Methods

The outcome measures set out below will be obtained prior to the intervention, at 15 sessions and 30 sessions, except for the sociodemographic and descriptive variables of the subject, which are reflected in Table 5.

### 2.11. Data Management and Statistical Analysis

Sample size: For instance, to achieve 80% power to detect differences in contrast to the null hypothesis H₀, mean difference in BESTest is equal to the limit of superiority, using a one-sided Student’s t-Test (superiority) for two independent samples, taking into account that the significance level is 5%, and assuming that the superiority limit is 7.81 units, as suggested in the article by Candance, et al., [55]. Additionally, the proportion of patients in the control group with respect to the total is 50%; it will be necessary to include 18 patients in the control group and 18 in the experimental group, a total of 36 patients in this study. Considering that the expected percentage of dropouts is 20%, it would be necessary to recruit 22 patients in the control group and 22 patients in the experimental group, a total of 44 patients in this study.

Patient characteristics and comparisons at baseline: To assess the comparability of the two groups, the characteristics of the patients assigned to each group will be presented in a table. Discrete variables will be summarized as frequencies and percentages. Continuous variables and time intervals will be summarized using means, standard deviation, or median and interquartile range. Both groups need to be homogeneous at baseline.

The statistical analysis will be carried out by an intention-to-treat analysis.

Analysis of the main objectives: Normality tests of the dependent variables were undertaken using the Kolmogorov–Smirnov test. The description of the results of the dependent variables was also performed, using either the mean and the standard deviation or the median and interquartile range, depending on the adjustment of these variables to normality. 

To respond to the main objective, with each primary result, hypothesis tests will be performed (where the alternative hypothesis is the greater efficacy of the experimental treatment compared to the conventional one), and estimates will be provided by 95% confidence intervals. In addition, the t-Student test will be used for independent samples in the parametric case and the Mann–Whitney U test for the non-parametric case.

Analysis of the secondary outcomes: For quality of life, contrasts of hypotheses will be made, and estimates will be provided by a 95% confidence interval. Furthermore, it is of interest to establish a regression model between quality of life and static balance, functional mobility, dynamic balance and gait in order to estimate their possible relationships. The strength of these relationships will be measured through the ANOVA F-test and the individual t-test. The strength of the relationship will be expressed as ORs and their 95% CI.

This project takes into account qualitative information to explore the reasons for possible dropouts through direct follow-up of patients.

## 3. Discussion

This study aims to investigate the effects of the application of Immersive VR combined with functional physiotherapy exercises on static and dynamic balance, and quality of life in patients with balance disorders secondary to stroke.

The most common alterations caused by stroke are associated with impairment of the motor components of movement, hemiparesis and balance disorders [56]. Static balance in these subjects is essential for activities such as standing, and dynamic balance for walking and climbing stairs. Therefore, a disturbance in either can compromise safe walking at home and in the community, increasing the likelihood of falls [57]. Both are indispensable for patients who have suffered a stroke because it is a key determinant of their quality of life [58].

There are clinical trials whose results [19,59,60,61,62] have shown that training based on VR systems is generally more effective than conventional treatment in relearning and improving balance, mobility and gait in patients post-stroke. In the same way, a systematic review report that the application of physiotherapy associated with the use of VR for patients who have suffered cerebrovascular accidents seems to be beneficial for static (measured by the BBS) and dynamic balance as a global result [18]. 

Currently, the most widely used VR systems to address motor symptoms in patients post-stroke are semi-immersive or non-immersive systems [63]. In addition, these non-immersive video game systems developed for home entertainment are less expensive, which has made this modality more accessible for possible rehabilitation interventions in patients post-stroke [64].

Technological advances have made it possible to start using Immersive VR as a therapeutic approach to improve motor function in stroke, although most studies focus on improving upper extremity function and self-care skills in patients post-stroke [65].

Given the small number of studies that use Immersive VR [18] in the treatment of balance in stroke subjects, more studies with larger samples and unified instruments to measure balance are needed.

The strong point of this study is the use of Immersive VR applied to the treatment of static and dynamic balance in the hospital environment. In this way, the use of Immersive VR systems, fully focusing the user’s attention on the task with the different scenarios, could have great advantages in the recovery of deficits. The greater the immersion in the virtual environment and the less contact with the outside physical world, the user’s attention and adherence to treatment will be enhanced. [66,67]. This is important since the most important cognitive alterations after stroke are problems of attention, concentration, visual processing, language, memory, reasoning language, memory, reasoning, problem-solving and higher executive functions, and that can have a negative impact on the learning of the subject with stroke [68]. In addition, the use of VR in the rehabilitation of the stroke patient has a positive impact on the acquisition of visual and spatial skills, which also increases the motivation and participation of the patient. [69].

The subject with stroke will work on static and dynamic balance immersed in real situations, although in a contained environment such as the hospital treatment unit. It should be noted that the findings of this work will mean a change in the treatment modality of people who have suffered a stroke.

The main limitation of this study was the impossibility of blinding the physiotherapists who performed the intervention. However, this situation is difficult to avoid in many physiotherapy treatments and, in this case, even due to the nature of the aim of this study.

The results of the trial will allow the technological incorporation of Immersive VR into physiotherapeutic protocols in the approach to stroke and will facilitate decision-making for the treatment of balance in people with this affectation.

Future research and clinical applications of VR should focus on more immersive systems [23].

## 4. Conclusions

The results of this study may add new insights into how to address balance using Immersive VR as part of physiotherapy treatments after stroke. In addition, if the new training approach proves effective, the results may provide insight into how to design more comprehensive protocols in the future for people with balance impairments after stroke.

## Figures and Tables

**Figure 1 healthcare-11-01335-f001:**
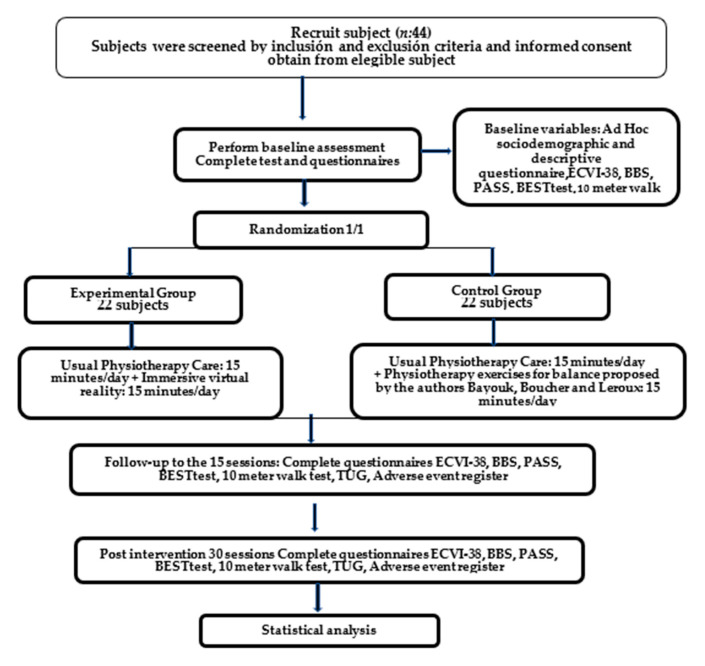
The study flowchart.

**Figure 2 healthcare-11-01335-f002:**
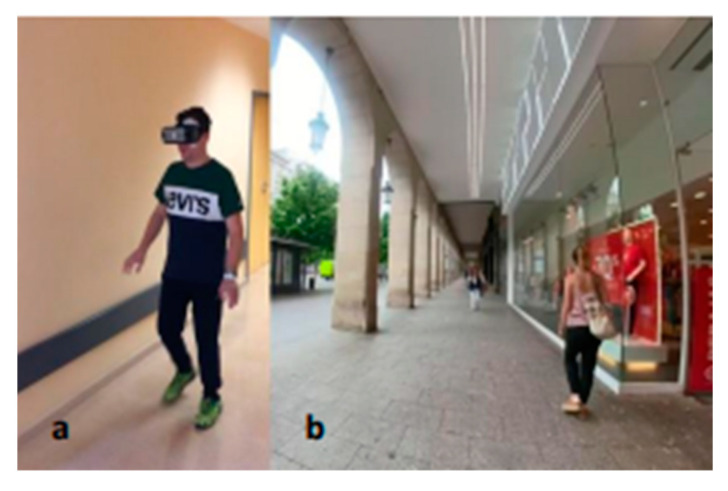
Two scenarios for static and dynamic balance training with VR. (**a**)*:* patient using FisioVR device, and (**b**): scenario that patient visualizes.

**Figure 3 healthcare-11-01335-f003:**
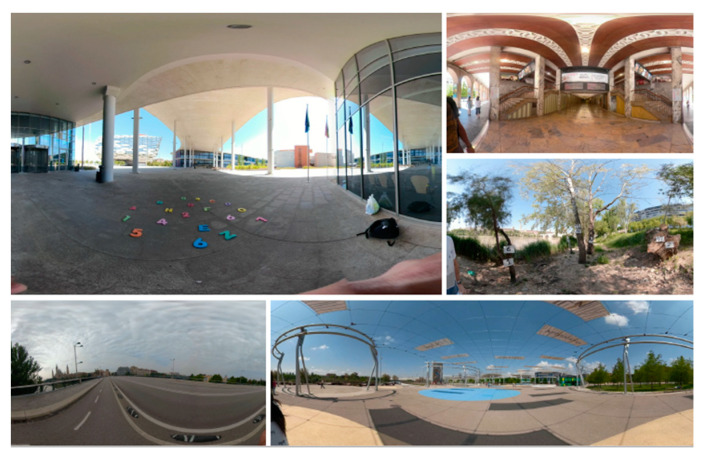
Immersive Virtual Reality scenarios.

**Table 1 healthcare-11-01335-t001:** Protocol of intervention.

Intervention Protocol	
Control Group (CG)	Intervention Group (IG)
15 min: physiotherapy treatment aimed at achieving functional improvement and increased postural control [44].	15 min: physiotherapy treatment aimed at achieving functional improvement and increased postural control [44].
15 min: static and dynamic balance training in sitting and standing according to Bayouk, Boucher and Leroux protocol [39].	15 min: static and dynamic balance training in sitting and standing through Immersive VR.

**Table 2 healthcare-11-01335-t002:** Exercises of the protocol of Bayouk, Boucher and Leroux (control group).

Exercises	Repetitions	Total Time
Stand with feet together for 10 s.	8 repetitions	2 min
Hold the tandem posture for 10 s.	8 repetitions	2 min
Get up from the chair without using arms.	3 sets of 8 repetitions	3 min
Walk forwards and backwards with one foot in front of the other.		5 min
Stay in monopodal support for 10 s.	10 repetitions	3 min

**Table 3 healthcare-11-01335-t003:** Outcomes measures.

Outcome Domain	Measures Instrument	T0	T1	T2
Sociodemographic and descriptive data	Ad Hoc questionnaire	X		
Primary outcomes				
Static balance and functional mobility	PASS	X	X	X
	BESTtest	X	X	X
Dynamic balance and gait	10 m walk test	X	X	X
	TUG	X	X	X
Secondary outcomes				
Quality of life	ECVI-38	X	X	X
Adverse effects	Open questions		X	X

T0 baseline, T1 evaluation at 15 sessions, T2 evaluation at 30 sessions. PASS- Postural Scale for Stroke Patients; TUG- Timed Up and Go; BESTest- Balance Evaluation System Test; ECVI-38- Quality of life scale for stroke.

**Table 4 healthcare-11-01335-t004:** Study period: Schedule of enrollment, interventions and assessments of this study.

STUDY PERIOD					
	Enrolment	Allocation	Post-Allocation		Closeout
Time points baseline	t	0	t15	t30	tx
Enrollment					
Eligibility screen	X				
Informed consent	X				
Allocation		X			
Assessments					
Baseline variables		X			
Post. Intervention and follow-up variables			X	X	X
Interventions					
IVR training group			X	X	
Physical therapy with Bayouk, Boucher and Leroux exercises			X	X	

IVR: Virtual Reality intervention group.

**Table 5 healthcare-11-01335-t005:** Collection of sociodemographic and descriptive data.

**Sociodemographic and descriptive data**	Age	Years
	Gender	Male/female
	BMI	Underweight/normal weight/overweight
	Hypertension	Yes/no
	Hypercholesterolemia	Yes /no
	Diabetes mellitus	Yes /no
	Tabaco habit	Yes /no
	Habitual consumption of stimulants or toxic substances	Yes /no
	Chronic drug use	Yes /no
	Mississippi Aphasia Screening Score	Score
	Mini-mental State Examination Score	Score
	Days of hospitalization	Number of days

BMI: Body Mass Index.

## Data Availability

Not applicable.

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
