# Peer review of "Effect of Physiotherapy Treatment with Immersive Virtual Reality in Subjects with Stroke: A Protocol for a Randomized Controlled Trial"

_healthcare, 2023, doi:10.3390/healthcare11091335_

Round 1

Author Response

Thank you for your contributions to the protocol:

  Effect of Physiotherapy treatment with immersive virtual reality in subjects with stroke

- Regarding how the secondary objectives are measured, they are presented in Table 3. Quality of life is measured with the ECVI-38 and adverse effects through open questions. It has not been wanted to put in this section of objectives so as not to duplicate content.

- Thank you for your contributions that have been included in the protocol with the text in blue

Thank you for giving us the opportunity to improve the protocol for publication in the Healthcare journal.

Reviewer 2 Report

Reviewer's comment

In this planned randomized controlled trial (RCT), the authors will investigate the quantity of virtual reality (VR) training. While several researchers developed VR-based training, the least effective training duration remained to be elucidated. In particular, there are many training menus for post-stroke patients to be implemented, while the total training time was limited to medical economics and patients' endurance capacity. In some cases, there is a shortage of VR devices in institutes or hospitals; thus, there may be a need to share a few numbers of devices with many patients. The authors' group plan to apply Bayouk, Boucher, and Leroux (BBL) exercises regarding static balance in sitting and standing, dynamic balance and quality of life as the control. Therefore, this study does not entirely meet the aim to reveal the excellent time duration, but it is more like investigating the alternativity of BBL exercise over VR training. Overall, the authors consider that this study may add new insight into how to address balance using immersive VR after a stroke, and the reviewer considers the study to meet this purpose.

Since it is a publication of RCT protocol, and the reviewer does not grasp their situation regarding setting, working pattern, available apparatus, or patient characteristics in practice, the reviewer proposes some concerns.

Concerns;

The pathogenesis of balance dysfunction is not considered.

Please consider including the diseases to affect balance function in the Exclusion criteria, such as vestibulocochlear disease, spinal cord injury or disease, and systemic peripheral neuropathy.

This study lacks observational assessment of stroke impairment like Somatosensory evoked potentials (SEP) or Motor evoked potentials (MEP). In particular, SEP can assess the function of the posterior funiculus, which closely relates to balance; therefore, this will propose further insight into the mechanisms. Moreover, SEP is applied in a stroke neurorehabilitation study.

Because blinding is difficult, the reviewer is concerned about how the authors will ensure the quality of BBL exercise in the control group.

Author Response

In this planned randomized controlled trial (RCT), the authors will investigate the quantity of virtual reality (VR) training. While several researchers developed VR-based training, the least effective training duration remained to be elucidated. In particular, there are many training menus for post-stroke patients to be implemented, while the total training time was limited to medical economics and patients' endurance capacity. In some cases, there is a shortage of VR devices in institutes or hospitals; thus, there may be a need to share a few numbers of devices with many patients. The authors' group plan to apply Bayouk, Boucher, and Leroux (BBL) exercises regarding static balance in sitting and standing, dynamic balance and quality of life as the control. Therefore, this study does not entirely meet the aim to reveal the excellent time duration, but it is more like investigating the alternativity of BBL exercise over VR training. Overall, the authors consider that this study may add new insight into how to address balance using immersive VR after a stroke, and the reviewer considers the study to meet this purpose.

-Comment 1. Since it is a publication of RCT protocol, and the reviewer does not grasp their situation regarding setting, working pattern, available apparatus, or patient characteristics in practice, the reviewer proposes some concerns.

Response 1. Thank you, we proceed to answer/include/modify the manuscript according to the reviewer's instructions. You will see the orange color changes to the text.

-Comment 2. The pathogenesis of balance dysfunction is not considered.

Response 2.Thanks for the contribution, we proceed to include in the introduction the phrase:

The alteration of balance in stroke that occurs after having suffered brain injuries that mainly affect the parieto-temporal or parieto-vestibular areas, which can cause a significant problem for the independence of these patients, since they present loss of the sense of touch , of the protective reactions and of the proprioceptive sense correlated with the ability to balance, as a consequence it will affect mobility, and it is the main consequence of falls in subjects with Ictus.

-Comment 3. Please consider including the diseases to affect balance function in the Exclusion criteria, such as vestibulocochlear disease, spinal cord injury or disease, and systemic peripheral neuropathy.

 Response 3. Thank you, we include pathologies that cause balance disorders but are not related to stroke in the exclusion criteria.

- vestibulocochlear disease, spinal cord injury or disease, systemic peripheral neuropathy, persistent phobic postural dizziness, migraine, Ramsay Hunt Syndrome and psychiatric disorders

- Comment 4.This study lacks observational assessment of stroke impairment like Somatosensory evoked potentials (SEP) or Motor evoked potentials (MEP). In particular, SEP can assess the function of the posterior funiculus, which closely relates to balance; therefore, this will propose further insight into the mechanisms. Moreover, SEP is applied in a stroke neurorehabilitation study.

Response 4. Thanks for the contribution, although in this study the SEP or MEP cannot be obtained to determine the deterioration in the cerebrovascular accident 

-Comment 5. Because blinding is difficult, the reviewer is concerned about how the authors will ensure the quality of BBL exercise in the control group.

Response 5. You're right, blinding is hard. Although the quality of the exercise program is the same for both groups, since all physiotherapists are trained on the management of these in subjects with stroke.

Thank you for your suggestion.

Reviewer 3 Report

Suggestions to improve:

a) In the introduction, a more detailed approach to the possible advantages from the point of view of body functions in the use of immersive virtual reality.

b) Not being the most important in the document, the time available for some phases of the exercises in phase 2 seems to be inaccurate: for example 10 repetitions of standing with feet together for 10s, is equal to 100s, leaving 20 seconds for the 'breaks'.

c) Include data on minimal clinically important difference and/or standard error of measurement for each instrument, if available.

d) When defining the sample size, justify which measure serves as a reference for the difference of 5 units.

e) In data analysis, explicitly state whether it will be performed on an intention-to-treat basis.

f) Given that in the intervention there is a common area in both groups, I suggest that the authors explore other statistical approaches in order to verify whether there are also improvements in the outcome indicators defined in the control group.

g) Given the expected high number of dropouts, it is suggested to collect additional information (perhaps qualitative) on the reasons for this.

Author Response

Thanks for the contributions to the manuscript to increase its quality. modifications are made in green on the manuscript

Suggestions to improve:

Comment 1. In the introduction, a more detailed approach to the possible advantages from the point of view of body functions in the use of immersive virtual reality.

Response 1. Thank you very much for the suggestion, we have implemented a more detailed approximation of the advantages from the point of view of bodily functions.

Remaining in the text in this way:

There are studies that demonstrate the effectiveness of the application of this immersive technology in older adult patients, related to cognitive conditions, in which older subjects have moderate emotional learning and show greater confidence in multisensory integration for learning 23,26–28, in Parkinson's disease 24,29–31 and on the balance, mobility and fatigue in multiple sclerosis 25,32. and it is promising to use gamified patient-tailored immersive VR for neglect rehabilitation after a stroke 33, it has also been seen that neuronal activity increases after the intervention, particularly in brain areas involving activate mirror neurons, as in the primary motor cortex 23,27,28,34

Comment 2. Not being the most important in the document, the time available for some phases of the exercises in phase 2 seems to be inaccurate: for example 10 repetitions of standing with feet together for 10s, is equal to 100s, leaving 20 seconds for the 'breaks'.

Response 2. You are right, it has been modified in the text.

Comment 3. Include data on minimal clinically important difference and/or standard error of measurement for each instrument, if available.

Response 3. Thank you, we have included data on the minimal clinically important difference for each instrument.

Comment 4. When defining the sample size, justify which measure serves as a reference for the difference of 5 units.

Response 4. Thanks for the question, the Bestest is going to be considered as the variable to use, it is answered in the manuscript as follows:

Sample size: For instance, to achieve 80% power to detect differences in contrast to the null hypothesis H₀: Mean difference in Bestest is equal to the limit of superiority, using a one-sided Student's T-Test (superiority) for two independent samples , taking into account that the significance level is 5%, and assuming that the superiority limit is 7.81 units, as suggested in the article by Candance et al. (   )   , and the proportion of pacients in the control group with respect to the total is 50%, it will be necessary to include 18 patients in the control group and 18 in the experimental group, totaling 36 patients in the study. Considering that the expected percentage of dropouts is 20%, it would be necessary to recruit 22 patients in the control group and 22 patients in the experimental group, a total of 44 patients in the study.

Patient characteristics and comparisons at baseline: To assess the comparability of the two groups, the characteristics of the patients assigned to each group will be presented in a table. Discrete variables will be summarized as frequencies and percentages. Continuous variables and time intervals will be summarized using means and standard deviation or median and interquartile range. Both groups need to be homogeneous at baseline.

The statistical analysis will be carried out by an intention-to-treat analysis.

Analysis of the main objectives: : Normality tests of the dependent variables were undertaken using the Kolmogorov-Smirnov. The description of the results of the dependent variables was also performed, using either the mean and the standard deviation or the median and interquartile range depending on the adjustment of these variables to normality.

To respond to main objective, with each primary result, hypothesis tests will be performed (where the alternative hypothesis is the greater efficacy of the experimental treatment compared to the conventional one) and estimates will be provided by 95% confidence intervals. The t-Student test will be used for independent samples in the parametric case and the Mann-Whitney U test for the non-parametric case.

Analysis of the secondary outcomes: For quality of life, contrasts of hypotheses will be made and estimates will be provided by 95% confidence interval. Furthermore, it is of interest to establish a regression model between quality of life and static balance, functional mobility, dynamic balance, and gait in order to estimate their possible relationships. The strength of these relationships will be measured through the ANOVA F-test and the individual t-test. The strength of the relationship will be expressed as ORs and their 95%CI.

Comment 5. In data analysis, explicitly state whether it will be performed on an intention-to-treat basis.

Response 5 Yes, it will be done by intention to treat. We added it to the manuscript.

Comments 6. Given that in the intervention there is a common area in both groups, I suggest that the authors explore other statistical approaches in order to verify whether there are also improvements in the outcome indicators defined in the control group.

Response 6.

The aim of this paper is to assess whether the IVR training program is better than exercise training. The reviewer's suggestions for this study are not considered because we have used in the control group the gold standard in balance work.

Comments 7. Given the expected high number of dropouts, it is suggested to collect additional information (perhaps qualitative) on the reasons for this.

Response 7.This project takes into account qualitative information to explore the reasons for possible dropouts, through direct follow-up of patients. It is included in this way in the paper.

Round 2

Reviewer 2 Report

The reviewer considers the draft is adequately revised.

Author Response

Thanks for the reviewer suggestions and giving us the opportunity to have the protocol published.
